# Regression from Upper One-side Labeled Data

## Abstract

We address a regression problem from weakly labeled data that are correctly labeled only above a regression line, i.e., upper one-side labeled data. The label values of the data are the results of sensing the magnitude of some phenomenon. In this case, the labels often contain missing or incomplete observations whose values are lower than those of correct observations and are also usually lower than the regression line. It follows that data labeled with lower values than the estimations of a regression function (lower-side data) are mixed with data that should originally be labeled above the regression line (upper-side data). When such missing label observations are observed in a non-negligible amount, we thus should assume our lower-side data to be unlabeled data that are a mix of original upper- and lower-side data. We formulate a regression problem from these upper-side labeled and lower-side unlabeled data. We then derive a learning algorithm in an unbiased and consistent manner to ordinary regression that is learned from data labeled correctly in both upper- and lower-side cases. Our key idea is that we can derive a gradient that requires only upper-side data and unlabeled data as the equivalent expression of that for ordinary regression. We additionally found that a specific class of losses enables us to learn unbiased solutions practically. In numerical experiments on synthetic and real-world datasets, we demonstrate the advantages of our algorithm.

## 1 Introduction

This paper addresses a scenario in which a regression function is learned for label sensor values that are the results of sensing the magnitude of some phenomenon. A lower sensor value means not only a relatively lower magnitude than a higher value but also a *missing or incomplete observation* of a monitored phenomenon. Label sensor values for missing observations are lower than those for when observations are correct without missing observations and are also usually lower than an optimal regression line that is learned from the correct observations. A naive regression algorithm using such labels causes the results of prediction to be low and is thus biased and underfitted in comparison with the optimal regression line.

In particular, when the data coverage of a label sensor is insufficient, the effect of missing observations causing there to be bias is critical. One practical example is that, for comfort in healthcare, we mimic and replace an intrusive wrist sensor (label sensor) with non-intrusive bed sensors (explanatory sensors). We learn a regression function that predicts the values of the wrist sensor from values of the bed sensors. The wrist sensor is wrapped around a wrist. It accurately represents the motion intensity of a person and is used such as for sleep-wake discrimination Tryon (2013); Mullaney et al. (1980); Webster et al. (1982); Cole et al. (1992). However, it can sense motion only on the forearm, which causes data coverage to be insufficient and observations of movements on other body parts to be missing frequently. The bed sensors are installed under a bed; while their accuracy is limited because of their non-intrusiveness, they have much broader data coverage than that of the wrist sensor. In this case, the wrist sensor values for missing observations are improperly low and also inconsistent with the bed sensor values as shown in Fig. 1-(1). This leads to severe bias and underfitting.

The specific problem causing the bias stems from the fact that our data labeled with lower values than the estimations of the regression function are mixed with data that should be originally labeled above the regression line. Here, we call data labeled above the regression line *upper-side* data, depicted as circles in Fig. 1-(2), and data labeled below the regression line *lower-side* data, depicted as squares in Fig. 1-(2). When there are missing observations, that is, our scenario, it means that the original data with missing observations have been moved to the lower side, depicted as triangles in Fig. 1-(3). We

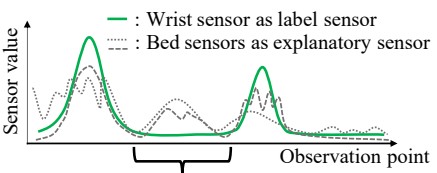
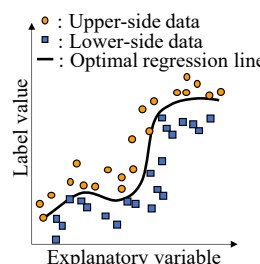
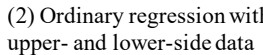
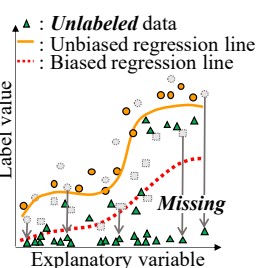

(1) Example application: we mimic wrist sensor having insufficient data coverage with bed sensors having broader data coverage.

(2) Ordinary regression with upper- and lower-side data

(3) Our setting with upper one-side data

Figure 1: One-side regression problem, where, due to missing observations, data are correctly labeled only above regression line, i.e., upper one-side. Regression function must be learned in unbiased and consistent manner to ordinary regression, where data are labeled correctly in both upper- and lower-side.

cannot determine which data have been moved by just examining the label values. It follows that our lower-side data are mixed with the original upper- and lower-side data.

We thus should assume our lower-side data to be *unlabeled* data, that is, a mix of original upper- and lower-side data. We overcome the bias by handling this asymmetric label corruption, in which upper-side data are correctly labeled but lower-side data are always unlabeled. There is an established approach against such corrupted *weak* labels in regression, that is, robust regression that regards weak labels as containing outliers Huber et al. (1964); Narula & Wellington (1982); Draper & Smith (1998); Wilcox (1997). However, since not asymmetric but rather symmetric label corruption is assumed there, it is still biased in our problem setting. In the classification problem setting, asymmetric label corruption is addressed with positive-unlabeled (PU) learning, where it is assumed that negative data cannot be obtained but unlabeled data are available as well as positive data Denis (1998); De Comité et al. (1999); Letouzey et al. (2000); Shi et al. (2018); Kato et al. (2019); Sakai & Shimizu (2019); Charoenphakdee & Sugiyama (2019); Li et al. (2019); Zhang et al. (2019); Xu et al. (2019); Zhang et al. (2020); Guo et al. (2020); Chen et al. (2020). The focus is on classification tasks, and an unbiased risk estimator has been proposed Du Plessis et al. (2014; 2015). There is a gap between the classification problem setting and our regression problem setting, i.e., we have to estimate specific continuous values, not positive/negative classes. We fill the gap with a novel approach for deriving an unbiased solution for our regression setting.

In this paper, we formulate a regression problem from upper one-side labeled data, in which the upper-side data are correctly labeled, and we regard lower-side data as unlabeled data. We refer to this as *one-side regression*. Using these upper-side labeled and lower-side unlabeled data, we derive a learning algorithm in an unbiased and consistent manner to ordinary regression that uses data labeled correctly in both upper- and lower-side cases. This is achieved by deriving our *gradient* that requires only upper-side data and unlabeled data as an asymptotically equivalent expression of that for ordinary regression. This is a key difference from the derivation of unbiased PU classification where *loss* has been used. We additionally found that a specific class of losses enables us to make it so that an unbiased solution can be learned practically. For implementing the algorithm, we propose a stochastic optimization method. In numerical experiments using synthetic and real-world datasets, we empirically evaluated the effectiveness of the proposed algorithm. We found that it improves performance against regression algorithms that assume that both upper- and lower-side data are correctly labeled.

## 2    ONE-SIDE REGRESSION

Our goal is to derive a learning algorithm with upper one-side labeled data in an unbiased and consistent manner to ordinary regression that uses both upper- and lower-side labeled data. We first

consider the ordinary regression problem; after that, we formulate a one-side regression problem by transforming the objective function of the ordinary one.

## 2.1 ORDINARY REGRESSION PROBLEM

Let $\boldsymbol{x} \in \mathbb{R}^D (D \in \mathbb{N})$ be a $D$-dimensional explanatory variable and $y \in \mathbb{R}$ be a real-valued label. We learn a regression function $f(\boldsymbol{x})$ that computes the value of an estimation of a label, $\hat{y}$, for a newly observed $\boldsymbol{x}$ as $\hat{y} = f(\boldsymbol{x})$. The optimal regression function $f^*$ is given by

$$f^* \equiv \underset{f}{\operatorname{argmin}} \, \mathcal{L}(f), \tag{1}$$

where $\mathcal{L}(f)$ is the expected loss when the regression function $f(\boldsymbol{x})$ is applied to data, $\boldsymbol{x}$ and $y$, distributed in accordance with an underlying probability distribution $p(\boldsymbol{x}, y)$:

$$\mathcal{L}(f) \equiv \mathbb{E}[L(f(\boldsymbol{x}), y)], \tag{2}$$

where $\mathbb{E}$ denotes the expectation over $p(\boldsymbol{x}, y)$, and $L(f(\boldsymbol{x}), y)$ is the loss function between $f(\boldsymbol{x})$ and $y$, e.g., the squared loss, $L(f(\boldsymbol{x}), y) = \|f(\boldsymbol{x}) - y\|_2^2$.

$\mathcal{L}(f)$ can be written by using the decomposed expectations $\mathbb{E}_{\text{up}}$ when labels are higher than estimations of the regression function ($f(\boldsymbol{x}) < y$, upper-side case) and $\mathbb{E}_{\text{lo}}$ when labels are lower than the estimations of the regression function ($y < f(\boldsymbol{x})$, lower-side case) as

$$\mathcal{L}(f) = \pi_{\text{up}}\mathbb{E}_{\text{up}}[L(f(\boldsymbol{x}), y)] + \pi_{\text{lo}}\mathbb{E}_{\text{lo}}[L(f(\boldsymbol{x}), y)], \tag{3}$$

where $\pi_{\text{up}}$ and $\pi_{\text{lo}}$ are the ratios for upper- and lower-side cases, respectively.

Note that the decomposition in Eq. (3) holds for any $f$ including $f^*$, and we omitted the decomposed expectation when $y = f(\boldsymbol{x})$ because it is always zero.

## 2.2 ONE-SIDE REGRESSION PROBLEM

We here consider a scenario in which we have training data, $\mathcal{D} \equiv \{\boldsymbol{x}_n, y_n\}_{n=1}^N$, that are correctly labeled only in the upper-side case because of the existence of missing label observations. The data in the lower-side case are a mix of original upper- and lower-side data and are considered to be unlabeled data. We can divide $\mathcal{D}$ by estimations of the regression function $f$ into upper-side data $\{\boldsymbol{X}_{\text{up}}, \boldsymbol{y}_{\text{up}}\} \equiv \{\boldsymbol{x}, y \in \mathcal{D} \mid f(\boldsymbol{x}) < y\}$ and unlabeled data $\boldsymbol{X}_{\text{un}} \equiv \{\boldsymbol{x} \in \mathcal{D} \mid y < f(\boldsymbol{x})\}$. In the ordinary regression, where both upper- and lower-side data are correctly labeled for training, expectations $\mathbb{E}_{\text{up}}$ and $\mathbb{E}_{\text{lo}}$ in Eq. (3) can be estimated by using the corresponding sample averages. In our setting, however, correctly labeled data from the lower-side case are unavailable, and, therefore, $\mathbb{E}_{\text{lo}}$ cannot be estimated directly.

We can avoid this problem by expressing $\mathcal{L}(f)$ as

$$\tilde{\mathcal{L}}(f) \equiv \pi_{\text{up}}\mathbb{E}_{\text{up}}[L(f(\boldsymbol{x}), y)] + \mathbb{E}[L(f(\boldsymbol{x}), \tilde{y}_{\text{lo}})] - \pi_{\text{up}}\mathbb{E}_{\text{up}}[L(f(\boldsymbol{x}), \tilde{y}_{\text{lo}})], \tag{4}$$

where expectation $\mathbb{E}$ for $\boldsymbol{x}$ can be estimated by computing a sample average for our unlabeled data $\boldsymbol{X}_{\text{un}}$, and $\tilde{y}_{\text{lo}}$ is a *virtual label* that is always lower than the estimations of the regression function $f(\boldsymbol{x})$, whose details will be given in the next paragraph. For this expression, the expected loss $\tilde{\mathcal{L}}(f)$ is represented by only the expectations over the upper-side data and unlabeled data, $\mathbb{E}_{\text{up}}$ and $\mathbb{E}$. Thus, we can design a gradient-based learning algorithm by using our training data. This transformation comes from Eqs. (2) and (3) with $\tilde{y}_{\text{lo}}$ as

$$\mathbb{E}[L(f(\boldsymbol{x}), \tilde{y}_{\text{lo}})] = \pi_{\text{up}}\mathbb{E}_{\text{up}}[L(f(\boldsymbol{x}), \tilde{y}_{\text{lo}})] + \pi_{\text{lo}}\mathbb{E}_{\text{lo}}[L(f(\boldsymbol{x}), \tilde{y}_{\text{lo}})]$$
$$\pi_{\text{lo}}\mathbb{E}_{\text{lo}}[L(f(\boldsymbol{x}), \tilde{y}_{\text{lo}})] = \mathbb{E}[L(f(\boldsymbol{x}), \tilde{y}_{\text{lo}})] - \pi_{\text{up}}\mathbb{E}_{\text{up}}[L(f(\boldsymbol{x}), \tilde{y}_{\text{lo}})]. \tag{5}$$

In practice, we cannot properly set the value of $\tilde{y}_{\text{lo}}$ as being always lower than $f(\boldsymbol{x})$. However, for learning based on gradients, this is not needed when we set the loss function as losses whose gradients do not depend on the value of $\tilde{y}_{\text{lo}}$ but just on the *sign* of $f(\boldsymbol{x}) - \tilde{y}_{\text{lo}}$, which is *always positive and* $\operatorname{sgn}(f(\boldsymbol{x}) - \tilde{y}_{\text{lo}}) = 1$ from the definition of $\tilde{y}_{\text{lo}}$, i.e., the loss functions satisfy

$$\frac{\partial L(f(\boldsymbol{x}), y)}{\partial \boldsymbol{\theta}} = g\big(\operatorname{sgn}(f(\boldsymbol{x}) - y), f(\boldsymbol{x})\big), \tag{6}$$

where $\boldsymbol{\theta}$ is the parameter vector of $f$, $g\big(\mathrm{sgn}(f(\boldsymbol{x}) - y), f(\boldsymbol{x})\big)$ is a gradient function depending on $\mathrm{sgn}(f(\boldsymbol{x}) - y)$ and $f(\boldsymbol{x})$, and $\mathrm{sgn}(\bullet)$ is a sign function. Common such losses are absolute loss and quantile losses. For example, the gradient of absolute loss, $|f(\boldsymbol{x}) - y|$, is

$$\frac{\partial |f(\boldsymbol{x}) - y|}{\partial \boldsymbol{\theta}} = \begin{cases} \frac{\partial f(\boldsymbol{x})}{\partial \boldsymbol{\theta}} & (\mathrm{sgn}(f(\boldsymbol{x}) - y) = 1) \\ -\frac{\partial f(\boldsymbol{x})}{\partial \boldsymbol{\theta}} & (\mathrm{sgn}(f(\boldsymbol{x}) - y) = -1) \\ \mathrm{Undefined} & (\mathrm{sgn}(f(\boldsymbol{x}) - y) = 0) \end{cases}, \tag{7}$$

which does not depend on the value of $y$ but just on the sign of $f(\boldsymbol{x}) - y$.

## 3 LEARNING WITH GRADIENT USING UPPER ONE-SIDE LABELED DATA

In this section, we derive the learning algorithm based on Eqs. (1) and (4) and show that it is unbiased to and consistent with ordinary regression. We consider the gradient of Eq. (4) by using losses that satisfy Eq. (6) for its second and third terms as

$$\frac{\partial \tilde{\mathcal{L}}(f)}{\partial \boldsymbol{\theta}} = \pi_{\mathrm{up}} \mathbb{E}_{\mathrm{up}} \left[ \frac{\partial L(f(\boldsymbol{x}), y)}{\partial \boldsymbol{\theta}} \right] + \mathbb{E}\big[g\big(\mathrm{sgn}(f(\boldsymbol{x}) - \tilde{y}_{\mathrm{lo}}), f(\boldsymbol{x})\big)\big]$$
$$- \pi_{\mathrm{up}} \mathbb{E}_{\mathrm{up}}\big[g\big(\mathrm{sgn}(f(\boldsymbol{x}) - \tilde{y}_{\mathrm{lo}}), f(\boldsymbol{x})\big)\big]. \tag{8}$$

Using upper-side and unlabeled sample sets, $\{\boldsymbol{X}_{\mathrm{up}}, \boldsymbol{y}_{\mathrm{up}}\}$ and $\boldsymbol{X}_{\mathrm{un}}$, the gradient in Eq. (8) can be estimated as

$$\frac{\partial \tilde{\mathcal{L}}(f)}{\partial \boldsymbol{\theta}} = \frac{\pi_{\mathrm{up}}}{n_{\mathrm{up}}} \left[ \sum_{\{\boldsymbol{x}, y\} \in \{\boldsymbol{X}_{\mathrm{up}}, \boldsymbol{y}_{\mathrm{up}}\}} \frac{\partial L(f(\boldsymbol{x}), y)}{\partial \boldsymbol{\theta}} \right] + \frac{1}{n_{\mathrm{un}}} \left[ \sum_{\boldsymbol{x} \in \boldsymbol{X}_{\mathrm{un}}} g\big(\mathrm{sgn}(f(\boldsymbol{x}) - \tilde{y}_{\mathrm{lo}}), f(\boldsymbol{x})\big) \right]$$
$$- \frac{\pi_{\mathrm{up}}}{n_{\mathrm{up}}} \left[ \sum_{\boldsymbol{x} \in \boldsymbol{X}_{\mathrm{up}}} g\big(\mathrm{sgn}(f(\boldsymbol{x}) - \tilde{y}_{\mathrm{lo}}), f(\boldsymbol{x})\big) \right], \tag{9}$$

where $\{\boldsymbol{x}, y\} \in \{\boldsymbol{X}_{\mathrm{up}}, \boldsymbol{y}_{\mathrm{up}}\}$ represent coupled pairs of $\boldsymbol{x}$ and $y$ in the upper-side sample set, and $n_{\mathrm{up}}$ and $n_{\mathrm{un}}$ are the numbers of samples for the upper-side and unlabeled sets, respectively.

By using the gradient in Eq. (9), we can optimize Eq. (1) and learn the regression function. Its unbiasedness and consistency will be given in Section 3.1, and the specific implementation of the algorithm will be given in Section 3.2.

### 3.1 UNBIASEDNESS AND CONSISTENCY OF GRADIENT

Our learning algorithm based on the gradient in Eq. (9) that uses only upper-side data and unlabeled data is justified as follows.

**Theorem 1.** *Suppose that loss function $L$ for the second term in Eq. (3) satisfies Eq. (6). Then, for any $f$, the gradient in Eq. (8) and its empirical approximation in Eq. (9) are unbiased to and consistent with the gradient of $\mathcal{L}(f)$ in Eq. (3).*

In other words, learning based on the gradient of Eq. (9), which uses only upper-side data and unlabeled data (one-side regression), asymptotically produces the same result as learning based on the gradient of $\mathcal{L}(f)$ in Eq. (3), which uses both upper- and lower-side data (ordinary regression).

*Proof.* First, by substituting Eq. (5) into the second and third terms in Eq. (8),

$$\frac{\partial \tilde{\mathcal{L}}(f)}{\partial \boldsymbol{\theta}} = \pi_{\mathrm{up}} \mathbb{E}_{\mathrm{up}} \left[ \frac{\partial L(f(\boldsymbol{x}), y)}{\partial \boldsymbol{\theta}} \right] + \pi_{\mathrm{lo}} \mathbb{E}_{\mathrm{lo}}[g\big(\mathrm{sgn}(f(\boldsymbol{x}) - \tilde{y}_{\mathrm{lo}}), f(\boldsymbol{x})\big)]. \tag{10}$$

Then, from the definitions of $\tilde{y}_{\mathrm{lo}}$ and $\mathbb{E}_{\mathrm{lo}}$, both in which $y$ is always $y < f(\boldsymbol{x})$,

$$\mathbb{E}_{\mathrm{lo}}[g\big(\mathrm{sgn}(f(\boldsymbol{x}) - \tilde{y}_{\mathrm{lo}}), f(\boldsymbol{x})\big)] = \mathbb{E}_{\mathrm{lo}}[g\big(1, f(\boldsymbol{x})\big)]$$
$$= \mathbb{E}_{\mathrm{lo}}[g\big(\mathrm{sgn}(f(\boldsymbol{x}) - y), f(\boldsymbol{x})\big)], \tag{11}$$

and, thus, the gradient (10) is essentially the same as the following gradient of the loss $\mathcal{L}(f)$ in Eq. (3) for ordinary regression when we set the loss function for the second term in Eq. (3) as losses that satisfy Eq. (6),

$$\frac{\partial \mathcal{L}(f)}{\partial \boldsymbol{\theta}} = \pi_{\mathrm{up}} \mathbb{E}_{\mathrm{up}}\left[\frac{\partial L(f(\boldsymbol{x}), y)}{\partial \boldsymbol{\theta}}\right] + \pi_{\mathrm{lo}} \mathbb{E}_{\mathrm{lo}}[g(\mathrm{sgn}(f(\boldsymbol{x}) - y), f(\boldsymbol{x}))]. \tag{12}$$

The gradient in Eq. (9) is also unbiased to and consistent with the gradient in Eq. (12), and its convergence rate is of the order $\mathcal{O}_p(1/\sqrt{n_{\mathrm{up}}} + 1/\sqrt{n_{\mathrm{un}}})$ in accordance with the central limit theorem Chung (1968), where $\mathcal{O}_p$ denotes the order in probability. □

## 3.2 Implementation of Learning Algorithm Based on Stochastic Optimization

We scale our algorithm based on Eq. (9) up by stochastic approximation with $M$-mini-batches and add a regularization term, $R(f)$:

$$\frac{\partial \tilde{\mathcal{L}}(f)}{\partial \boldsymbol{\theta}} = \sum_{m=1}^{M}\left[\left[\sum_{\{\boldsymbol{x},y\}\in\left\{\boldsymbol{X}_{\mathrm{up}}^{\{m\}}, \boldsymbol{y}_{\mathrm{up}}^{\{m\}}\right\}} \frac{\partial L(f(\boldsymbol{x}), y)}{\partial \boldsymbol{\theta}}\right] + \rho\left[\sum_{\boldsymbol{x}\in\boldsymbol{X}_{\mathrm{un}}^{\{m\}}} g(1, f(\boldsymbol{x}))\right]\right.$$
$$\left. - \left[\sum_{\boldsymbol{x}\in\boldsymbol{X}_{\mathrm{up}}^{\{m\}}} g(1, f(\boldsymbol{x}))\right]\right] + \lambda\frac{\partial R(f)}{\partial \boldsymbol{\theta}}, \tag{13}$$

where $\{\boldsymbol{X}_{\mathrm{up}}^{\{m\}}, \boldsymbol{y}_{\mathrm{up}}^{\{m\}}\}$ and $\boldsymbol{X}_{\mathrm{un}}^{\{m\}}$ are upper-side and unlabeled sets in the $m$-th mini-batch, respectively, $\lambda$ is a regularization parameter, and the regularization term $R(f)$ is, for example, the L1 or L2 norm for the parameters $\boldsymbol{\theta}$. We also convert $n_{\mathrm{up}}/(\pi_{\mathrm{up}}n_{\mathrm{un}})$ as $\rho$ ignoring constant coefficients and apply $\mathrm{sgn}(f(\boldsymbol{x}) - \tilde{y}_{\mathrm{lo}}) = 1$. The hyperparameters $\rho$ and $\lambda$ are optimized in training.

We can learn the regression function with the gradient in Eq. (13) by using any stochastic gradient method, such as Adam Kingma & Ba (2015) and FOBOS Duchi & Singer (2009). The algorithm is described in Algorithm 1. In the following experiments, we used Adam with the hyperparameters recommended in Kingma & Ba (2015), and the number of samples in the mini-batches was set to 32. By using the learned $f(\boldsymbol{x})$, we can estimate $\hat{y} = f(\boldsymbol{x})$ for new data.

From a practical perspective, the first term in Eqs. (9) and (13) requires that the estimations for the upper-side samples be higher than their label values as much as possible because $\sum_{\{\boldsymbol{x},y\}\in\{\boldsymbol{X}_{\mathrm{up}}, \boldsymbol{y}_{\mathrm{up}}\}} L(f(\boldsymbol{x}), y)$ becomes zero when every estimation of the regression function $f(\boldsymbol{x})$ is located in $y \leq f(\boldsymbol{x})$. In contrast, the second term requires that the estimations for unlabeled samples be just as small as possible. The third term balances the effect of the second term.

Our algorithm and discussions are applicable to a scenario that is opposite the one-side case, where the data are correctly labeled only on the lower side. Since the derivation is obvious from the analogy of the upper one-side case, we just show its learning algorithm for the lower one-side case in the supplementary material. One example of the lower one-side case is when the label sensor has ideal coverage, but the cost of observation is high, and we need to mimic sensor values with other cheaper sensors having smaller coverage.

## 4 Experimental Results

We now empirically test the effectiveness of the proposed approach. Our goal is to investigate the impact of our unbiased gradient, which is derived from the objective function based on the assumption of upper one-side labeled data in Eq. (4). We thus show how the proposed method improves performance against regression methods whose objective functions assume that both upper- and lower-side data are correctly labeled. We use the same model and optimization method for all of the methods, and the only difference is their objective functions.

### 4.1 Experimental Setup and Datasets

We report the *mean absolute error* (MAE) and its standard error between the estimation results $\hat{\boldsymbol{y}} = \{\hat{y}_n\}_{n=1}^{N}$ and the corresponding true labels $\check{\boldsymbol{y}}$ across 5-fold cross-validation, each with a different

---

**Algorithm 1** One-side regression based on stochastic gradient method

---

**Input:** Training data $\mathcal{D} = \{\boldsymbol{x}_n, y_n\}_{n=1}^N$ and hyperparameters $\rho, \lambda \geq 0$
**Output:** Model parameters $\boldsymbol{\theta}$ for $f$
 1: Let $\mathcal{A}$ be an external stochastic gradient method and $G_m$ be a gradient for the $m$-th mini-batch
 2: **while** No stopping criterion has been met
 3:   Shuffle $\mathcal{D}$ into $M$-mini-batches, and denote by $\{\boldsymbol{X}^{\{m\}}, \boldsymbol{y}^{\{m\}}\}$ the $m$-th mini-batch whose size is $N_m$
 4:   **for** $m = 1$ **to** $M$
 5:     $G_m \leftarrow 0$
 6:     **for** $n = 1$ **to** $N_m$
 7:       **if** $f\left(\boldsymbol{x}_n^{\{m\}}\right) - y_n^{\{m\}} < 0$ **then**
 8:         $G_m \leftarrow G_m + \frac{\partial L\left(f\left(\boldsymbol{x}_n^{\{m\}}\right), y_n^{\{m\}}\right)}{\partial \boldsymbol{\theta}} - g\left(1, f\left(\boldsymbol{x}_n^{\{m\}}\right)\right) + \lambda \frac{\partial R(f)}{\partial \boldsymbol{\theta}}$
 9:       **else**
10:         $G_m \leftarrow G_m + \rho g\left(1, f\left(\boldsymbol{x}_n^{\{m\}}\right)\right) + \lambda \frac{\partial R(f)}{\partial \boldsymbol{\theta}}$
11:       Update $\boldsymbol{\theta}$ by $\mathcal{A}$ with $G_m$

---

randomly sampled training-testing split. MAE is defined as $\text{MAE}(\check{\boldsymbol{y}}, \hat{\boldsymbol{y}}) = 1/N \sum_{n=1}^N |\check{y}_n - \hat{y}_n|$. For each fold of the cross-validation, we used a randomly sampled 20% of the training set as a validation set to choose the best hyperparameters for each algorithm, where hyperparameters providing the highest MAE in the validation set were chosen. All of the experiments were carried out with a Python implementation on workstations having 48-80 GB of memory and 2.3-4.0 GHz CPUs. With this environment, the computational time was a few hours for producing the results for each dataset.

**Data** 1**: Synthetic dataset.** Using a synthetic dataset, we investigated whether our algorithm could indeed learn from upper one-side labeled data. We randomly generated $N$ training samples, $\boldsymbol{X} = \{\boldsymbol{x}_n\}_{n=1}^N$, from the standard Gaussian distribution $\mathcal{N}(\boldsymbol{x}_n; 0, \boldsymbol{I})$, where the number of samples was $N = 1,000$, the number of features in $\boldsymbol{x}$ was $D = 10$, and $\boldsymbol{I}$ is the identity matrix. Then, using $\boldsymbol{X}$, we generated the corresponding $N$ sets of true labels $\check{\boldsymbol{y}} = \{\check{y}_n\}_{n=1}^N$ from the distribution $\mathcal{N}(\check{y}_n; \boldsymbol{w}^\top \boldsymbol{x}_n, \beta)$, where $\boldsymbol{w}$ are coefficients that were also randomly generated from the standard Gaussian distribution $\mathcal{N}(\boldsymbol{w}; 0, \boldsymbol{I})$, $\beta$ is the noise precision, and $\top$ denotes the transpose. For simulating the situation in which a label sensor has missing observations, we created training labels $\boldsymbol{y} = \{y_n\}_{n=1}^N$ by randomly selecting $K$ percent of data in $\check{\boldsymbol{y}}$ and replacing their values with the minimum value of $\check{\boldsymbol{y}}$. We finally added white Gaussian noise whose precision was the same as that of $\check{\boldsymbol{y}}$ for the replaced $K$ percent of data. We repeatedly evaluated the proposed method for each of the following settings. The noise precision was $\beta = \{10^0, 10^{-1}\}$, which corresponded to low- and high-noise settings, and the proportion of missing training samples was $K = \{25, 50, 75\}\%$. In the case of $K = 75\%$, only 25 percent of the samples correctly corresponded to labels, and all of the other samples were attached with labels that were lower than the corresponding true values. In general, it is quite hard to learn regression functions using such data. In the experiment on Data 1, we used a linear model, $\boldsymbol{\theta}^\top \boldsymbol{x}$, for $f(\boldsymbol{x})$ and an implementation for Eq. (13) with squared loss for the first term, absolute loss, which satisfies Eq. (6), for the second and third terms, and L1-regularization for the regularization term. Loss functions having such a heterogeneous aspect are often used in the literature, e.g., Huber loss Huber et al. (1964), Epsilon-insensitive loss Vapnik (1995), and quantile losses Koenker & Bassett Jr (1978). We set the candidates of the hyperparameters, $\rho$ and $\lambda$, to $\{10^{-3}, 10^{-2}, 10^{-1}, 10^0\}$. We standardized the data by subtracting their mean and dividing by their standard deviation in the training split.

**Data** 2**: Kaggle dataset with synthetic corruption.** We used a real-world sensor dataset collected from the Kaggle dataset Sen (2016) that contains breathing signals. For this dataset, we used signals from a chest belt as $\boldsymbol{X} = \{\boldsymbol{x}_n\}_{n=1}^N$ and signals obtained by the Douglas bag (DB) method, which is the gold standard for measuring ventilation, as true labels $\check{\boldsymbol{y}} = \{\check{y}_n\}_{n=1}^N$. The dataset consisted of $N = 1,432$ samples, and $\boldsymbol{x}$ in each sample had $D = 2$ number of features, i.e., the period and height of the expansion/contraction of the chest. For our problem setting, we created training labels $\boldsymbol{y} = \{y_n\}_{n=1}^N$ by randomly selecting $K$ percent of data in $\check{\boldsymbol{y}}$ and replacing their value with the minimum value of $\check{\boldsymbol{y}}$. We finally added white Gaussian noise whose standard deviation was $0.1 \times s$ for the replaced $K$ percent of data, where $s$ is the standard deviation of the original $\check{\boldsymbol{y}}$. The setting for $K$ was the same as that of Data 1, $K = \{25, 50, 75\}\%$. In the experiment on Data 2, for its non-

Table 1: Comparison of proposed method and methods based on various objective functions in terms of MAE (smaller is better). We show best methods in bold.

(1) Data 1: Synthetic dataset

| | Low-noise setting ($\beta = 10^0$) | | | High-noise setting ($\beta = 10^{-1}$) | | |
|---|---|---|---|---|---|---|
| | $K = 25\%$ | $K = 50\%$ | $K = 75\%$ | $K = 25\%$ | $K = 50\%$ | $K = 75\%$ |
| MSE | $0.77\pm0.01$ | $1.53\pm0.02$ | $2.30\pm0.02$ | $1.03\pm0.02$ | $1.62\pm0.03$ | $2.36\pm0.03$ |
| Proposed | $\mathbf{0.58\pm0.01}$ | $\mathbf{0.60\pm0.01}$ | $\mathbf{0.60\pm0.01}$ | $\mathbf{0.78\pm0.02}$ | $\mathbf{0.79\pm0.02}$ | $\mathbf{0.79\pm0.02}$ |

(2) Data 2: Kaggle dataset with synthetic corruption

| | $K = 25\%$ | $K = 50\%$ | $K = 75\%$ |
|---|---|---|---|
| MSE | $0.55\pm0.02$ | $0.91\pm0.02$ | $1.32\pm0.02$ |
| Proposed | $\mathbf{0.43\pm0.01}$ | $\mathbf{0.46\pm0.01}$ | $\mathbf{0.59\pm0.01}$ |

(3) Data 3: Real-world UCI dataset

| | Class A | Class B | Class C | Class D | Class E | Avg. |
|---|---|---|---|---|---|---|
| MSE | $2.38\pm0.03$ | $1.54\pm0.01$ | $1.42\pm0.01$ | $1.37\pm0.01$ | $1.21\pm0.01$ | $1.58$ |
| MAE | $2.14\pm0.02$ | $1.46\pm0.01$ | $1.44\pm0.01$ | $1.33\pm0.01$ | $1.31\pm0.01$ | $1.54$ |
| Huber | $2.04\pm0.02$ | $1.66\pm0.01$ | $1.45\pm0.01$ | $1.50\pm0.01$ | $1.32\pm0.01$ | $1.59$ |
| Proposed-1 | $1.55\pm0.02$ | $1.18\pm0.01$ | $1.11\pm0.01$ | $1.14\pm0.01$ | $1.03\pm0.01$ | $1.20$ |
| Proposed-2 | $\mathbf{1.32\pm0.01}$ | $\mathbf{0.99\pm0.01}$ | $\mathbf{0.94\pm0.01}$ | $\mathbf{0.86\pm0.01}$ | $\mathbf{0.97\pm0.01}$ | $\mathbf{1.02}$ |

linearity, we used $\boldsymbol{\theta}^\top \boldsymbol{\phi}(\boldsymbol{x}, \sigma)$ for $f(\boldsymbol{x})$, where $\phi$ is a radial basis function, and $\sigma$ is a hyperparameter representing the kernel width that is also optimized in the training split. We set the candidates of the hyperparameters, $\rho$, $\lambda$, and $\sigma$, to $\{10^{-3}, 10^{-2}, 10^{-1}, 10^0\}$. The other implementation was the same as that for Data 1.

**Data 3: Real-world UCI dataset.** We here applied the algorithm to a real sensor dataset, which was collected from the UCI Machine Learning Repository Velloso (2013); Velloso et al. (2013). It contains sensor outputs from dumbbells and from wearable devices attached to the arm, forearm, and waist during exercise. We used all of the features from the dumbbell sensor that took "None" values less than ten times as $\boldsymbol{X} = \{\boldsymbol{x}_n\}_{n=1}^N$, where each sample had $D = 13$ number of features. We used the magnitude of acceleration on the arm as training labels $\boldsymbol{y} = \{y_n\}_{n=1}^N$, which had insufficient data coverage and missing observations for the movements of other body parts. For testing, we used the magnitude of acceleration for the entire body as true labels $\tilde{\boldsymbol{y}} = \{\tilde{y}_n\}_{n=1}^N$. Because there were five classes for the exercise task with severe mode changes between classes, we divided the dataset into five datasets on the basis of class: A ($N = 11,159$), B ($N = 7,593$), C ($N = 6,844$), D ($N = 6,432$), and E ($N = 7,214$). In the experiment on Data 3, we used a 6-layer multilayer perceptron with ReLU Nair & Hinton (2010) (more specifically, $D$-100-100-100-100-1) as $f(\boldsymbol{x})$ in order to demonstrate the usefulness of the proposed method in training deep neural networks. We also used a dropout Srivastava et al. (2014) with a rate of $50\%$ after each fully connected layer. We used two implementations for the first term in Eq. (13) with absolute loss (Proposed-1) and squared loss (Proposed-2). For both implementations, we used the absolute loss, which satisfies Eq. (6), for the second and third terms and used L1-regularization for the regularization term. We set the candidates of the hyperparameters, $\rho$ and $\lambda$, to $\{10^{-3}, 10^{-2}, 10^{-1}, 10^0\}$. The other implementation was the same as that for Data 1.

## 4.2 PERFORMANCE COMPARISON

Table 1-(1) and -(2) show the performance on Data 1 and Data 2 for the proposed method and an ordinary regression method that uses *mean squared error* (MSE) assuming that both upper- and lower-side data are correctly labeled as its objective function. This comparison shows whether our method could learn from upper one-side labeled data, from which the ordinary regression method could not learn. From Table 1-(1) and -(2), we can see that the overall performance of the proposed method was significantly better than that of MSE. We found that the performance of our method was not significantly affected by the increase in the proportion of missing training samples $K$ even for $K = 75\%$, unlike that of MSE. Table 1-(3) shows a more extensive comparison using the real-world UCI dataset (Data 3) between our methods, Proposed-1 and Proposed-2, and methods based on

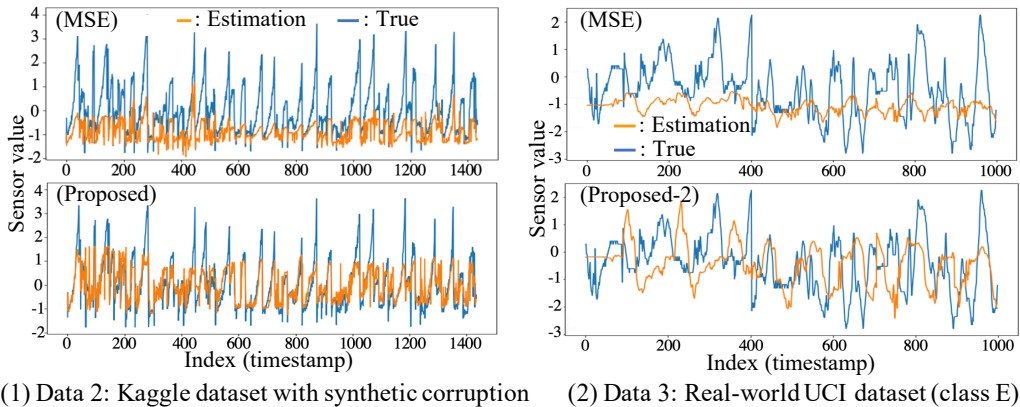

(1) Data 2: Kaggle dataset with synthetic corruption     (2) Data 3: Real-world UCI dataset (class E)

Figure 2: Comparison of estimation results of proposed method and those of MSE. Orange line represents estimation results of each method, and blue line represents true label values.

various objective functions consisting of MSE, MAE, and Huber losses Huber et al. (1964); Narula & Wellington (1982); Wilcox (1997). The regression methods based on MAE and Huber losses were robust regression methods that assume symmetric label corruption. From Table 1-(3), we can see that the performance of Proposed-1 and Proposed-2 was totally better than that of the baselines. The robust regression methods did not improve in performance against MSE. In particular, Proposed-1 and Proposed-2 respectively reduced the error by more than $20\%$ and $30\%$ compared with the other methods on average.

**Demonstration of unbiased learning and prediction.** Figure 2-(1) compares the estimation results of the proposed method with true labels and those of MSE for the Kaggle dataset with synthetic corruption (Data 2). Since the ordinary regression method, MSE, regards both upper- and lower-side data as correctly labeled, we can see that it produced biased results due to the missing observations. The proposed method did not. Figure 2-(2) shows a comparison of the estimation results between the proposed method, Proposed-2, and MSE for the real-world UCI dataset (Data 3). For ease of viewing, we show the results for the first $1,000$ samples for the class E data, where the errors of most of the methods were the lowest. Although MSE showed the lowest error among the baselines for the class E data, we can see that the predictions by MSE were somewhat biased and underfitted for the real data having our assumed nature. This was not the case for the proposed method.

### 4.3 REAL HEALTHCARE CASE STUDY

We show the results of a healthcare case study in the supplementary material, where we estimated the motion intensity of a participant that was measured accurately with an intrusive sensor wrapped around the wrist (ActiGraph) Tryon (2013); Mullaney et al. (1980); Webster et al. (1982); Cole et al. (1992) from non-intrusive bed sensors that were installed under a bed. The results showed that the intrusive sensor could be replaced with the non-intrusive ones, which would be quite useful for reducing the burden on users.

### 5 CONCLUSION

We formulated a one-side regression problem using upper-side labeled and lower-side unlabeled data and proposed a learning algorithm for it. We showed that our learning algorithm is unbiased to and consistent with ordinary regression that uses data labeled correctly in both upper- and lower-side cases. We developed a stochastic optimization method for implementing the algorithm. An experimental evaluation using synthetic and real-world datasets demonstrated that the proposed algorithm was significantly better than regression algorithms without the assumption of upper one-side labeled data.

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
