# OpenReview forum: "Regression from Upper One-side Labeled Data"
_ICLR.cc/2021/Conference — Reject_

### Official Review · AnonReviewer2 · 2020-10-24
**New problem setting and algorithm with limited technical contributions**

**Rating:** 5
**Confidence:** 4

**Review:**

In this paper, the authors address a new weakly supervised regression problem. In this problem setting, upper-side data (labeled above the regression line) and unlabeled data are provided. To solve this problem, the authors derive a learning algorithm in an unbiased and
consistent manner to ordinary regression that is learned from data labeled correctly in both upper- and lower-side cases. Experiments demonstrate the advantages of the proposed algorithm.

Pros:
1.	To the best of my knowledge, this paper is the first to solve the weakly supervised regression problem presented in the paper. I consider that it is the biggest advantage of this paper.
2.	This paper proposes a consistent learning algorithm to solve the above problem.
3.	Experiments demonstrate the effectiveness of the proposed algorithm.

Cons:
1.	The presentation of this paper needs to be improved. For example, I understand that in the introduction section, the authors try to justify that the weakly supervised regression problem (where upper-side and unlabeled data are available) is reasonable and could be encountered in real-world settings. However, I personally feel that the presentation is not very clear and I am not fully convinced. In addition, for the order of Eq. (4) and Eq. (5), I think it would be better to present Eq. (5) before Eq. (4), as Eq. (4) relies on Eq. (5).
2.	For the proposed consistent algorithm, I would admit that it is novel to some degree, while the technical contribution of this algorithm is limited. It is worth noting that the proposed algorithm is adapted from the risk estimator of PU learning (Du Plessis et al., 2014; 2015). I think the only key contribution lies in Eq. (6), e.g., the authors show that instead of setting the value of ${\tilde{y}}\_{\text{lo}}$, we can find the gradient only depends on the sign of $f(\boldsymbol{x})-{\tilde{y}}\_{\text{lo}}$.
3.	For the experiments, it seems that the authors do not use a ground-truth regression line to separate the given data, and obtain upper-side and unlabeled data. Instead, they corrupt some selected data by setting their value to the minimum regression value. I feel that this practical operation does not accord with the proposed problem setting. Maybe we could use originally labeled data to obtain a well-trained regression line and then obtain the required upper-side and unlabeled data.

In summary, this paper proposed a novel problem setting and a novel learning algorithm, while the problem setting is not well justified and the technical contribution of the algorithm is limited.

---

> ### Author Response · Authors · 2020-11-17
> **Response to Reviewer 2**
>
> We thank for your careful reviews and valuable suggestions for our paper. We will update our paper based on your comments, such as the order of Eq. (4) and Eq. (5).
>
> [Experimental setting for synthetic data]
> It is impossible to obtain the upper-side and lower-side data exactly in real data. Thus, we conducted the experiments on synthetic data where we do not know which data should be upper-side or unlabeled (lower-side) to evaluate the feasibility of our method.

---

### Official Review · AnonReviewer1 · 2020-10-29
**Bypassing data corruption in upper one-side labeled data**

**Rating:** 4
**Confidence:** 4

**Review:**

The authors study the problem of training a regression model when only for a subset of the datapoints (those for which their label lie above the current model prediction) the correct labels are available. A few comments,

1) It is unclear if the labels y can be noisy. I assume they can't be, because all the derivations seem to be under the assumption they are not.
2) The application of this setup is not clear to me. I think the authors would benefit from motivating it via other papers that study the same regression problem (if any), as opposed to citing other topics in motion sensor research. They would also benefit from writing down the existing work on classification more explicitly, and connecting it to the regression setup in the paper. It is unclear if the classification literature treats a similar (or exactly the same) problem in the classification setting and what is the hard part of translating these results into regression.

The solution to the problem proposed by the authors is quite simple. This would not be a downside if the motivation of the problem and the related work was established with more authority at the beginning of the problem. I am concerned that in the case of losses of the form g(sgn(f(x)-y), f(x)) the problem is not meaningful because in this case the learner only requires to know if f(x) < y or not. The initial problem the authors set to solve vanishes in this case. This means that Theorem 1 is not very informative.

In section 3.2, there is little explanation as to why using a \rho multiplier in (13). This does not seem to be in accordance with Theorem 1. It is also unclear to me why do the last two terms on the RHS of (4) need to be written as a difference, when at the end gradients of an expectation of a loss of the form g(sgn(f(x)-y)), f(x)) over the unlabeled dataset can be computed directly. There doesn't seem to be any need of writing it as a difference.

The experimental evaluation is thorough.

---

> ### Author Response · Authors · 2020-11-17
> **Response to Reviewer 1**
>
> We thank for your careful reviews and valuable suggestions for our paper.
>
> [Gradient of the form g(sgn(f(x)-y) is meaningful and common]
> As mentioned in the last paragraph in Section2.2, gradients of the form g(sgn(f(x)-y) appear in such as the least absolute regression, which works well and is a common regression method.
>
> [Why do the last two terms on the RHS of (4) need to be written as a difference?]
> Because we do not have any lower-side data, we need to write the loss with only upper-side data and unlabeled data. We rewrite Eq.(3), which requires both upper-side data and lower-side data, into Eq.(4), which requires only upper-side data and unlabeled data.
>
> [y can be noisy]
> We assume the existence of the noise in the loss function in Eq.(2), which means expected loss over the corresponding distribution.

---

### Official Review · AnonReviewer5 · 2020-11-04
**Official Blind Review #5**

**Rating:** 5
**Confidence:** 5

**Review:**

Summary: This paper considers a regression setting in which the missing values are observed with lower values than the true values. Authors provided appealing application for this problem setting. They rewrote the risk and provided an unbiased gradient estimator. However, there is a gap between the estimator and the actual implementation, thus making the overall paper less convincible.

Main Concern:
- A gap exists between Eq. (8) and Eq. (13). In Eq. (8), the expectation is taken over the distribution of "up". This distribution, as well as \pi_{up}, is fixed throughout training. Unlike PU classification, this essential information is not given in this problem setting. However, according to Eq. (13) and Algorithm 1, this distribution and \pi_{up} change ever minibatch with the current model. I admit this is a pratical algorithm, but it differs substaintialy from the first half of the paper. To fill the gap, investigation on how Eq. (13) approximate Eq. (8) should be conducted at least.

---

> ### Author Response · Authors · 2020-11-17
> **Response to Reviewer 5**
>
> We thank for your careful reviews and valuable suggestions for our paper.
>
> [Justification for Eq.(13)]
> Eq.(13) is just a mini-batch approximation for the unbiased gradient in Eq.(9), it works well same to the ordinary mini-batch approximation. Also, we would like to note that the decomposition in Eq.(3) is for general regression problem not specifically ours. In Eq.(3), E_{up} and \pi_{up} are originally changing depending on f and it is not our assumption or proposal.
> As shown in Theorem 1, for any f, the gradients in Eq.(8) and Eq.(9) are unbiased to and consistent with the gradient of L(f) in Eq.(3). It also means that for any E_{up} and the corresponding distribution for upper-side case, the gradient in Eq.(9) is unbiased and consistent. Consequently, changing E_{up} and the corresponding upper-side samples for every updates in the gradient descent with the current model does not affect the theorem.

---

> > ### Comment · AnonReviewer5 · 2020-11-20
> > **Data should be generated from f*.**
> >
> > I admit that for Eq.(3), you can change for every possible function. However, data should be generated from an unknown underlying function f*. This means, one observed dataset corresponds to one single underlying function. Thus, when given a dataset, the underlying f* is accordingly fixed, thus the corresponding seperation for up data and lower data should also be fixed. Thereforer, I still consider changing this separation for each step is still an unjustified approximation.

---

> > > ### Author Response · Authors · 2020-11-24
> > > **Response to Reviewer 5**
> > >
> > > Thank you very much for your additional reviews and suggestions.
> > > In our formulation, E and p(x, y), which produce data, are fixed. From the definition of Eq.(3), E_{up} and E_{lo} depend on a current f for both ordinary regression and our one-side regression. As you know, f is changing in SGD and we can see that E_{up} and E_{lo} are also changing in each step of SGD from the view point of Eq.(3), but E and p(x, y) are not changing. Thus, since Theorem 1 holds for each step of SGD, we can say that Eq.(13) is a mini-batch approximation for the unbiased gradient in Eq.(9).
> > > Also, we justified the effectiveness of Eq.(13) and Algorithm 1 with experimental results, as the other reviewers mentioned. In the main text, we also have explicitly described that Eq.(13) is an approximation for the gradient in Eq.(9).
> > > The point of our manuscript is to show that we can derive a gradient for the one-side regression in an unbiased and consistent manner to that for ordinary regression, that is Theorem 1, and to show we can develop a practical algorithm to implement that in a straightforward approximation, a mini-batch approximation.

---

> > > > ### Comment · AnonReviewer5 · 2020-11-25
> > > > **Sample Approximation for Eq.(13)**
> > > >
> > > > Thank you for your further explanation. I agree the discussion is fine until Eq.(13). However, In Eq.(13), you need samples from the distribution E_{up} and E_{lo}. I think the key disagreement between me and the authors is that I consider these two distributions are not varying during training. For example, in Figure 1 (1), the missing observasion period is fixed when recording the data, and will not change when you learn using the observed data. Thus, choosing different samples by different f in Algorithm 1 line 7 to 10 is the same as using noisy samples to approximate the expectation over a distribution.

---

> > > > > ### Comment · AnonReviewer5 · 2020-11-25
> > > > > **Positive Response**
> > > > >
> > > > > On the other side, I admit contributions on empirical evaluation should be noted and raised my score. But I still think the paper need to be presented in an alternative way to be accepted, paying more attention on the empirical side.

---

### Decision · Program_Chairs · 2021-01-07
**Final Decision**

**Decision:**

Reject

**Comment:**

The paper addresses regression in a weakly supervised setting where the correct labels are only available for examples whose prediction lie above some threshold. The paper proposes a method using a gradient that is unbiased and consistent.

Pros:
- Problem setting is new and this paper is one of the first works exploring it.
- The procedure comes with some unbiasedness and consistency guarantees.
- Experimental results on a wide variety of datasets and domains.

Cons:
- Novelty and technical contribution is limited.
- Motivation of the problem setting was found to be unclear.
- Some gaps in the experimental section (i.e. needing the use of synthetic data or synthetic modifications of the real data).

Overall, the reviewers felt that as presented, the paper did not convincingly motivate the proposed upper one-sided regression problem as important or relevant in practice, which was a key reason for rejection. The paper may contain some nice ideas and I recommend taking the reviewer feedback to improve the presentation.